# Spectral decomposition unlocks ascidian morphogenesis

Joel Dokmegang[1,2]*, Emmanuel Faure[3,4], Patrick Lemaire[3,5], Edwin Munro[6], Madhav Mani[1,2]*

[1]Northwestern University, Evanston, United States; [2]NSF-Simons Center for Quantitative Biology, Evanston, United States; [3]University of Montpellier, Montpellier, France; [4]LIRMM, Montpellier, France; [5]CNRS, Paris, France; [6]University of Chicago, Chicago, United States

## eLife Assessment

In this **important** work, a quantitative analysis method for three-dimensional morphogenetic processes during embryonic development is introduced. The proposed method is a pipeline combining several methods, allowing quantitative analysis of developmental processes without cell segmentation and tracking. Upon application of their method, the authors obtain **convincing** evidence that ascidian gastrulation is a two-step process. This work should be of interest to a broad range of developmental biologists who aim to obtain a quantitative understanding of morphogenesis.

## Abstract

Describing morphogenesis generally consists in aggregating the multiple high-resolution spatiotemporal processes involved into reproducible low-dimensional morphological processes consistent across individuals of the same species or group. In order to achieve this goal, biologists often have to submit movies issued from live imaging of developing embryos either to a qualitative analysis or to basic statistical analysis. These approaches, however, present noticeable drawbacks as they can be time consuming, hence unfit for scale, and often lack standardization and a firm foundation. In this work, we leverage the power of a continuum mechanics approach and flexibility of spectral decompositions to propose a standardized framework for automatic detection and timing of morphological processes. First, we quantify whole-embryo scale shape changes in developing ascidian embryos by statistically estimating the strain rate tensor field of its time-evolving surface without the requirement of cellular segmentation and tracking. We then apply to this data spectral decomposition in space using spherical harmonics and in time using wavelets transforms. These transformations result in the identification of the principal dynamical modes of ascidian embryogenesis and the automatic unveiling of its blueprint in the form of scalograms that tell the story of development in ascidian embryos.

## Introduction

Morphogenesis, the emergence of shape in living systems, is a continuous process littered with spatiotemporal dynamics at various timescales and lengthscales and significance. Developmental biology aims at the identification, localization, and timing of these processes. Once this work is carried out in a given species, embryogenesis can then be described as a series of stages delineated in space and time by the identified landmarks (*Satoh, 1978*; *Nishida, 1986*; *Jeffery, 1992*; *Keller et al., 2003*; *Lemaire, 2009*; *Sherrard et al., 2010*; *Hashimoto et al., 2015*; *Hashimoto and Munro, 2018*; *Guignard, 2020*). In order to rigorously define development landmarks, biologists have mostly had to submit

*For correspondence:
jdokmegang@gmail.com (JD);
madhav.mani@northwestern.edu
(MM)

Competing interest: The authors declare that no competing interests exist.

imaged embryos either to qualitative analyses or rudimentary statistical analysis. These methods, however, present major drawbacks. On the one hand, they can be time consuming, hence unfit for scale. On the other hand, since morphogenetic processes tend to be unique to a species, these simple methods often lack a general language and framework that permit comparative analyses. For instance, whereas the analysis of cell counts can inform about the proliferation dynamics in a tissue, it does not reveal anything about the shape of the system. For this purpose, other measurements such as length, width, height, aspect ratios, or curvatures would be more suitable. Although efforts have been made to automate the staging of development in living systems (*Jones et al., 2022*), these methods still rely on preliminary examination using traditional methods.

A standardized method able to identify key milestones in development and lay out the blueprint of morphogenesis in a given system is henceforth needed. Recent breakthroughs in microscopy technology have propelled the resolution of live imaging data to the subcellular scale, allowing for the uncovering of precise cell and tissue shape dynamics (*Tassy et al., 2006*; *Stelzer, 2015*; *Power and Huisken, 2017*). These advances have created an unprecedented opportunity for the leveraging of computational methods in the study of morphogenesis (*Tassy et al., 2006*; *Michelin et al., 2015*; *Stegmaier et al., 2016*; *Leggio et al., 2019*; *Guignard, 2020*). The rigorous and physically motivated framework of continuum mechanics accommodates itself well to the flow-like dynamics of biological tissues (*Humphrey, 2003*; *Ambrosi et al., 2011*; *Blanchard et al., 2009*; *Humphrey, 2013*; *Streichan et al., 2018*). Within this framework, strain rate fields, which measure the rate at which the shape of a system changes with time, are suited to characterize the dynamical behavior of the system. Moreover, mounting evidence has informed of the requirement for embryo-wide approaches in the study of morphogenetic flows (*Streichan et al., 2018*; *Mitchell et al., 2022*). However, although the evaluation of such global fields across the spatial and temporal domains spanned by a system of interest may reveal valuable insights into its dynamical workings (*Bar-Kochba et al., 2015*; *Stout et al., 2016*; *Patel et al., 2018*), their sole determination might not be sufficient for a holistic description of the behavior of the system: there is a need for novel methods to analyze them.

This is especially true when it comes to morphogenesis (*Dalmasso et al., 2021*; *Romeo et al., 2021*; *Mitchell and Cislo, 2022*). The processes involved in development are inherently multiscale, both in the spatial and temporal domains, and may interact or overlap (*Godard and Heisenberg, 2019*; *Dokmegang et al., 2021*; *Dokmegang, 2022*). As is the case with several species (*Godard and Heisenberg, 2019*), ascidian early development is a playground featuring important displays of cellular divisions and tissue mechanics (*Lemaire, 2009*). The local behaviors captured by indicators such as the strain rate field might therefore arise from a non-trivial superposition of these dynamical modes, essentially making these measurements complex to interpret without further analysis. Spectral decomposition, whereby a signal is broken down into its canonical components, is well suited to the study of systems that exhibit multimodal behaviors (*Romeo et al., 2021*; *Dalmasso et al., 2021*). The benefits are at least twofold: (i) individual constituents may represent distinct dynamical processes, thereby enabling the decoupling of physical processes entangled in the data; and (ii) only a handful of components may significantly contribute to the original function, resulting in a compressed, lower dimensional representation that capture the main features of the studied process. The canonical components usually take the form of well-known families of functions whose linear combination can reconstitute the original field.

In this work, we take advantage of microscopy imaging data to develop a generic computational framework able to identify and delineate the main features of morphogenesis. Our method takes as input *3D+time* images of developing ascidian embryos and outputs spatiotemporal scalograms of ascidian development in the form of heatmaps that highlight key developmental processes and stages of ascidian gastrulation. By virtue of a novel meshing scheme derived from level-set methods, raw cell geometry data is first transformed into a single time-evolving embryonic surface on which the strain rate tensor field can be computed. The accuracy of our inference of a strain rate field relies on high-frequency temporal sampling, characterized by small deformations of the embryonic surface between subsequent time points. The morphomaps we present are a result of spectral analyses of the strain rate fields, featuring spherical harmonics decomposition in the spatial domain and wavelet decomposition in the temporal domain. In summary, our method can identify and classify dynamical morphogenetic events. In particular, we are able to identify and distinguish the morphogenetic modes of gastrulation and neurulation phases in ascidian

development, recover the characteristic two-step sequence of endoderm invagination originally described using 3D analysis of cell shapes (*Sherrard et al., 2010*), and capture patterns of cellular divisions in ascidian development (*Nishida, 1986*). Moreover, our method identifies a distinctive stage of ascidian gastrulation, *'blastophore closure'*, which follows endoderm invagination and precedes neurulation.

## Results

### Definition of Lagrangian markers on the surface of the embryo

In order to recover a continuum description of the dynamics in ascidian morphogenesis, we aim to examine the time evolution of strain rate fields across the entire surface of developing embryos. This endeavor, however, presents at least two significant challenges. On the one hand, strain rate computation requires the presence of fiducial markers on the surface of the embryo. Characteristically, this requirement is not always accounted for in the imaging of developing embryos. On the other hand, the outer layer of the embryo being constituted of single-cell apical faces, even if such markers had been defined at an earlier time point, uncontrolled stochastic biological processes such as proliferation within the tissue might subsequently grossly uneven the distribution of these markers, thus rendering the computed mechanical indicators at best imprecise. Given the non-triviality of an experimental setup able to solve the described issues, a computational method is required. The goal of such a method would be to computationally discretize an embryo surface into a set of material particles whose trajectories can be tracked in small lapses of development. The positions of these markers over time can then be used to derive mechanical indicators of development dynamics (*Humphrey, 2003*).

To achieve this fit, we first take advantage of the level set scheme described in *Zhao et al., 2000* to define static markers on the surface of the embryo at every timepoint of development. The gist of our method resides in the definition of a homeomorphic map between the surface of the embryo ($S_1(t)$) and a topologically equivalent mesh ($S_2(t)$) whose number of vertices, faces, and edges remain fixed (*Figure 1a*). Conformal parametrizations of embryonic shape have been used in other systems (*Alba et al., 2021*). Here, the topologically equivalent mesh consists of a sphere resulting from successive butterfly subdivisions of an icosahedron (*Hardy and Steeb, 2008*). Using a homeomorphic map, this mesh can be deformed to match the surface of the embryo at each time point of development (*Figure 1b*). As in *Zhao et al., 2000*, the map is obtained by finding the positions of $S_2(t)$ vertices that minimize the distance between both surfaces (*Figure 1a*, right). At the initial time point, $S_2(t)$ is chosen to be a sphere enclosing the embryo. For various reasons, including computational efficiency, variants of this method can be defined such that, for instance, at subsequent time steps, linear combinations of the sphere and its deformations matching the embryo at preceding steps are used. Further details are given in Appendix 1.

Next, via a numerical study, we assert that markers defined as such behave as Lagrangian particles in small increments of developmental dynamics. To support this point, we identify on the raw dataset the positions of cellular junctions at the surface of the embryo and evaluate how well our virtual markers mimic their movements in time. We measure the relative displacement between a cellular junction (*Figure 1c*, red points) and its corresponding vertex (*Figure 1c*, green points) at consecutive time points. We take the difference between these distances and normalize it by the average side length of cell apices. Despite gross approximations inherent to the nature of the dataset (geometric meshes) and the process of identifying cellular junctions (averaging the barycenters of closest triangles between three or more cells in contact), the relative displacement between cellular junctions and their markers remains on average relatively small (under 8%, *Figure 1d*). We further show that this characteristic is independent of the number of particles used in the method (*Figure 1c*), making it a remarkable property of the scheme. This result sheds even more favorable light on the method when considering that cellular junctions, precisely because they are the meeting point of three or more cells, are expected to exhibit more chaotic behavior than single-cell particles. Moreover, these numbers are skewed by large-scale morphogenesis processes such as synchronized cell divisions, as evidenced by the spikes in *Figure 1d*, and fast-paced endoderm gastrulation, as highlighted by the higher errors at the vegetal pole of the embryo during this phase (red dots in *Figure 1c*, right).

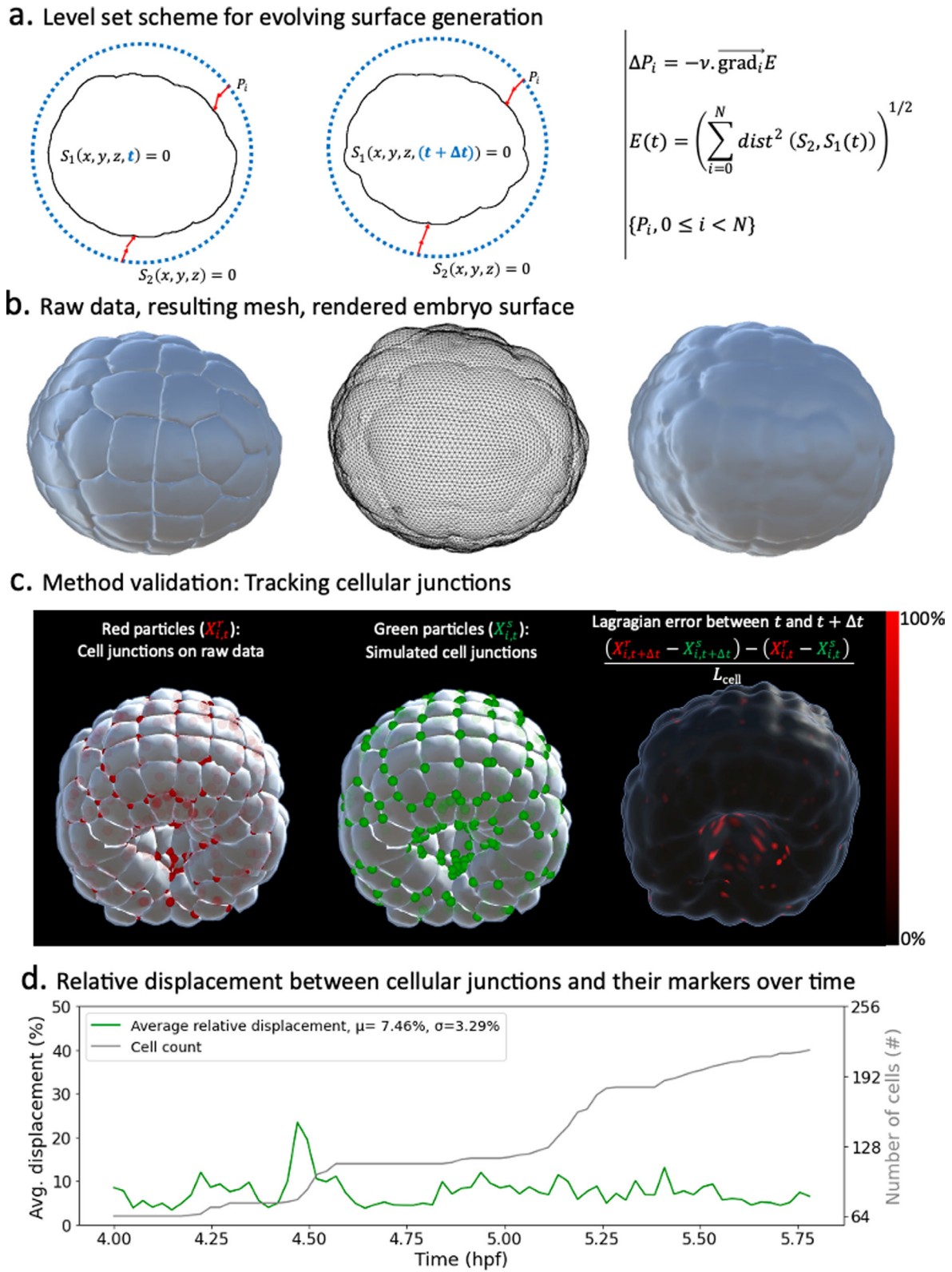

**Figure 1.** Level sets-inspired Lagrangian markers. (**a**) Left: schematics of the level set method. Right: fundamentals of the numerical scheme that shapes $S_2$ into $S_1$. (**b**) Illustration of the method in action. Left: raw data consisting of geometric meshes of single cells spatially organized into the embryo. Center: embryo surface mesh resulting from the application of the level set scheme. Right: rendering of the embryonic surface. (**c**) Tracking of cellular junctions. Left: identification of cellular junctions (red dots). Center: corresponding markers (green dots), defined as vertices on the computed

*Figure 1 continued on next page*

*Figure 1 continued*

embryonic surface closest to the junctions. Right: relative displacement between junctions and their markers at consecutive timepoints. (**d**) Plot over time of the relative displacement between cellular junctions and their markers.

## Strain rate field describes ascidian morphogenesis

Once a mesh representing the surface has been constructed for the embryo surface, we proceed with the computation of the strain rate fields across the surface of the embryo and throughout development timeline. Thanks to the Lagrangian nature of mesh vertices, a velocity field can be defined on the mesh. Although particles at every given time point live on the 2D surface of the embryo, their trajectories in time involve greater degrees of freedom in the 3D space. A correct parametrization of the velocity field at every position henceforth requires three coordinates $\mathbf{v}(\mathbf{x}) = (v_x(\mathbf{x}), v_y(\mathbf{x}), v_z(\mathbf{x}))^T$. The strain rate field is derived as the symmetric part of the discrete gradient of the velocity field, computed as described in *Mancinelli et al., 2018*. Intuitively, the strain rate evaluated on a given mesh vertex measures how the velocity vector varies in the neighborhood (*Mancinelli et al., 2018*; *De Goes et al., 2020*).

$$D(t, \mathbf{x}) = sym(\nabla \mathbf{v}(t, \mathbf{x})) \tag{1}$$

The mathematical construction of the strain rate (*Equation 1*) implies that its algebraic representation takes the form of a second-order tensor that can be written down as a $3 \times 3$ matrix $(D)_{ij}$. The diagonal elements of this matrix capture the linear strain rate in the $x, y, z$ axes, depicting the change in length per unit time. The non-diagonal elements stand for shear strain rates in the $xy, xz$, and $yz$ directions. Because $D$ is symmetric, there exists an alternative representation that holds stronger local geometric meaning. This representation is obtained by computing the eigenvectors and eigenvalues of the strain rate tensor. Eigenvectors stand for orthogonal spatial directions that are not rotated, but only stretched, by the application of the strain rate matrix. They define the principal axes of a coordinate system in which the strain rate tensor would be solely composed of maximal linear strain rates (*Figure 2a*). From this decomposition, we derive a scalar field that is computed at every mesh particle as the square root of the sum of the squared eigenvalues of the strain rate (*Figure 2b*). Intuitively, this field describes the magnitude of the rate of change underwent by a particle at the surface of the embryo in the three orthogonal spatial directions of most significant rate of change.

In order to minimize undesirable artifacts that may arise from numerical inefficiencies, we apply a Gaussian filter to the strain rate tensor field before deriving the scalar field. At each particle location, we apply a Gaussian convolution mask spanning its first- and second-order neighborhood. A similar smoothing process is also used in the time domain. Interestingly, this strain rate-derived scalar field remarkably mirrors well-known features of ascidian development. Similarities between the spatiotemporal distribution of morphogenesis processes described in the literature and heatmaps of this field on the evolving embryo surface emerge. On the one hand, wider spatial gradients of yellow to red depicting higher morphological activity portray the spatiotemporal locations of endoderm invagination in the embryonic vegetal pole (*Figure 2b*, center-left; *Sherrard et al., 2010*), synchronized rounds of division in the animal pole, and zippering in the neural plate (*Hashimoto et al., 2015*; *Figure 2b*, center-right, right). On the other hand, known spatiotemporal locations of low morphological significance (e.g., the animal pole when not proliferating) in the embryo exhibit stronger concentration of mechanical activity on cell boundaries, with the corollary that cellular identities are mostly preserved (*Figure 2b*, $t = 4$ hpb). A notable by-product of this scalar field is the evidencing of the duality of the embryo as both a sum of parts constituted of cells and an emerging entity in itself: the strain rate field clearly discriminates between spatiotemporal locations where isolated single-cell behaviors are preponderant (e.g., *Figure 2b*, $t = 4$ hpb) and those where coordinated cell behaviors dominate (e.g., *Figure 2b*, $t = 5.48$ hpb).

This brief overview already demonstrates the riches in a quantitative, spatially global and not event-driven approach to study morphogenesis. It also sets the stage for further analysis of morphogenesis dynamics in the ascidian embryo.

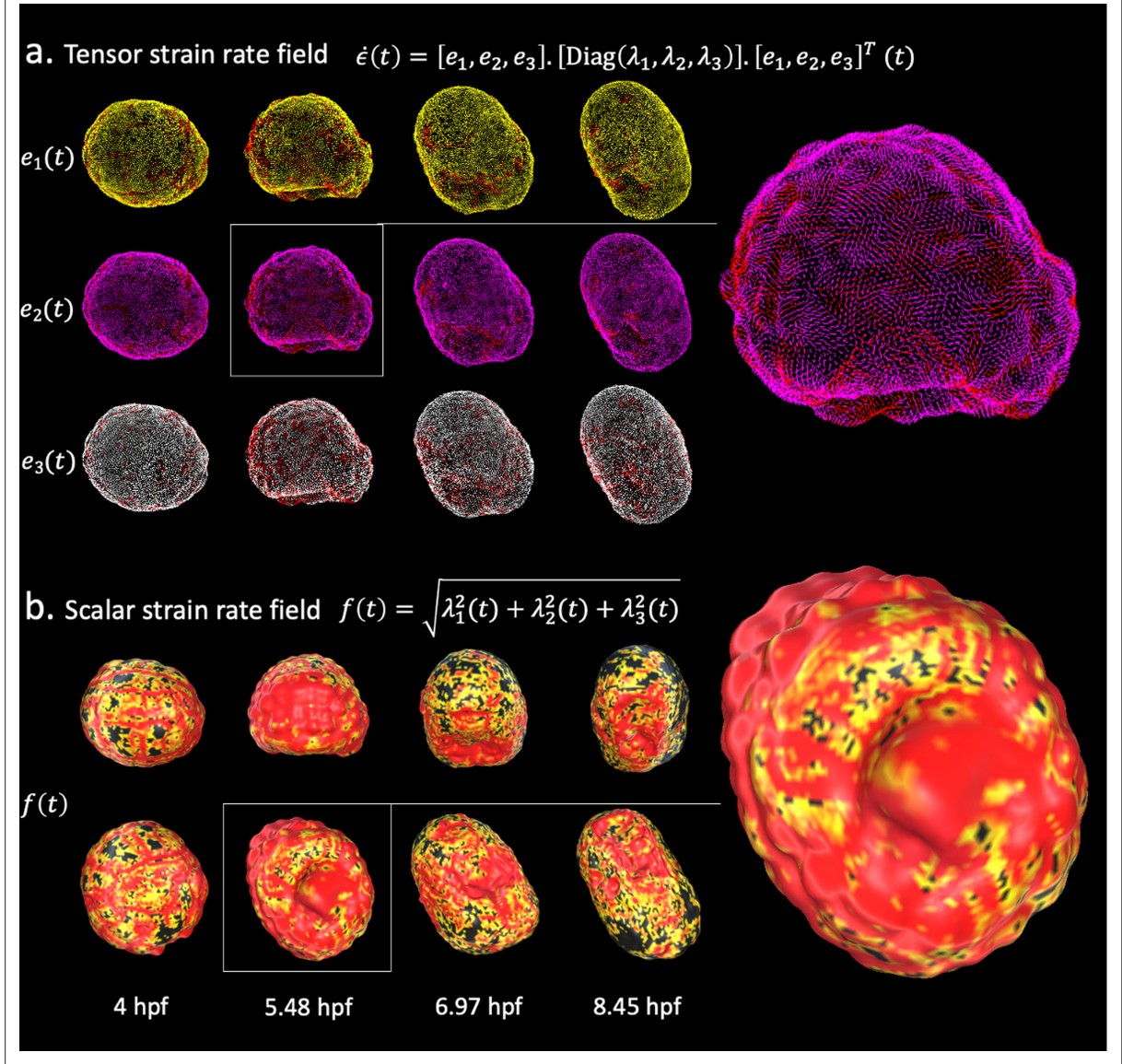

**Figure 2.** Strain rate field describes morphogenesis. The strain rate tensor field measures the rate at which morphological changes occur in the embryo as a function of time. The strain rate tensor field is locally represented as a 3 × 3 symmetric matrix and is completely determined by its eigenvector fields. (**a**) Heatmap of the eigenvector fields of the strain rate tensor. Each row represents a vector field distinguished by a distinct root color (yellow, pink, white). The gradient from the root color to red represents increasing magnitudes of the strain rate tensor. Top: spatiotemporal dynamics of the first eigenvector field. Middle: spatiotemporal dynamics of the second eigenvector field. Bottom: spatiotemporal dynamics of the third eigenvector field. (**b**) Heatmap of the scalar strain rate field. The gradient from yellow to red depicts regions of increasing morphological activity, while black stands for areas of low morphological activity. The heatmaps show high morphological activity in the invaginating endoderm and zippering neural plate, but also across the embryonic animal during rounds of synchronised division.

## Spectral decomposition in space: Spherical harmonics reveal the main modes of ascidian morphogenesis

In order to capture relevant features of the strain rate field in the spatial domain, we conduct a spectral analysis of the scalar strain rate field. The family of spherical harmonic functions stands out as a de facto standard for the study of signals defined on a unit sphere, and by extension on surfaces homeomorphic to the sphere. Spherical harmonics form an infinite orthonormal basis of functions defined on the surface of the sphere and represent a generalization of the Fourier series for functions of two variables (*Knaack and Stenflo, 2005*). Unsurprisingly, these functions play an important role in many branches of science displaying spherical symmetry, including quantum mechanics and geophysics

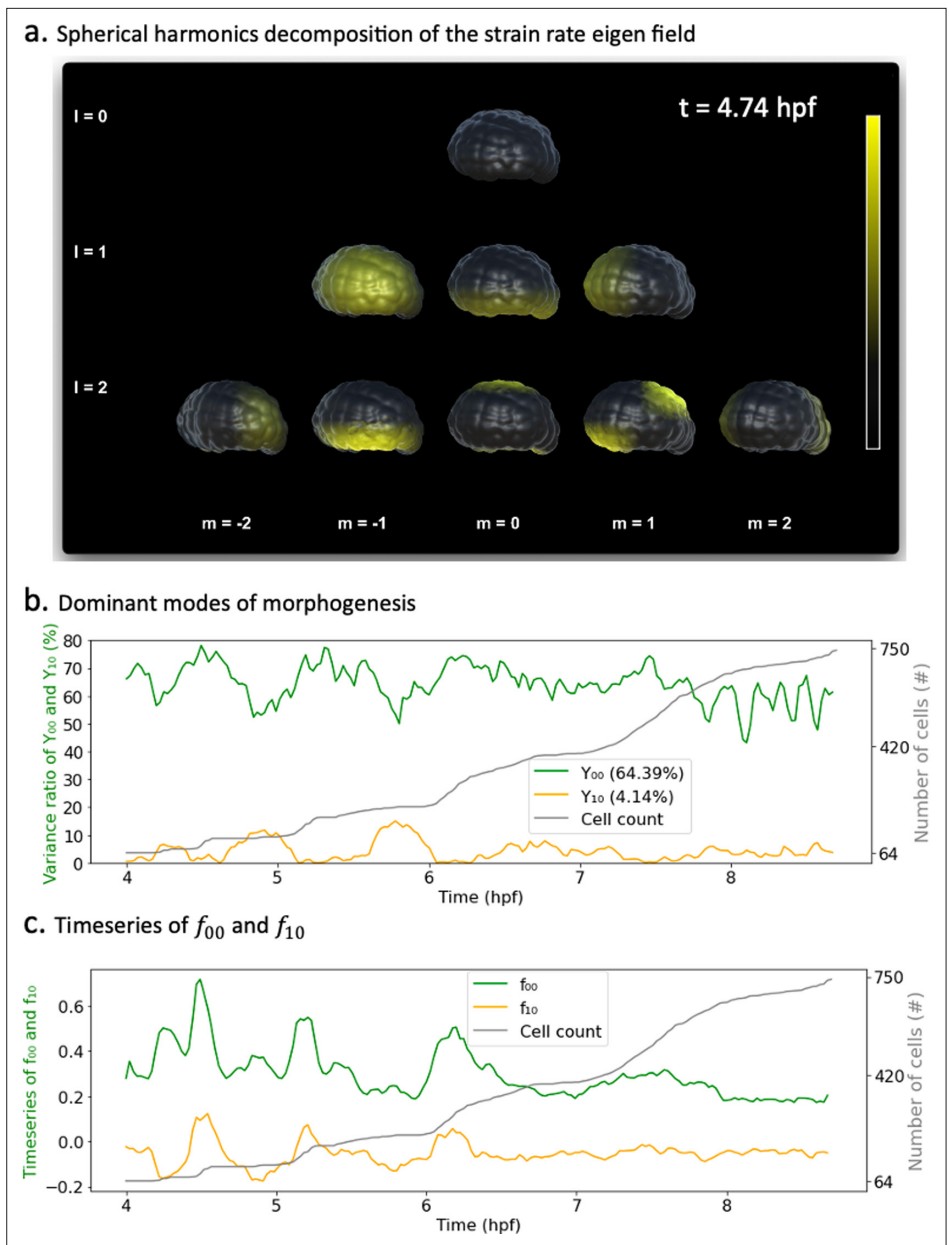

**Figure 3.** Spherical harmonics decomposition of morphogenesis. (**a**) Example of spherical harmonics decomposition of the scalar strain rate field mapped to the embryo at $t = 4.74$ hpf. Each picture represents the value of the harmonic field ($Y_{lm}$) multiplied by its coefficient $f_{lm}$. $f_{lm}s$ here are taken relative to each specific $f_{lm}$ minimal and maximum bounds in the entire time window of observation. Thresholding is applied for better rendering. (**b**) Time evolution of the variance ratios of the main modes of ascidian early morphogenesis ($Y_{00}$ and $Y_{10}$). The cell population dynamic is also included in the plot for clarity. (**c**) Time evolution of the coefficients $f_{00}$ and $f_{10}$ associated with spherical harmonics ($Y_{00}$ and $Y_{10}$). The cell population count is also included in the plot for clarity.

(**Knaack and Stenflo, 2005**; **Dahlen and Tromp, 2021**). Spherical harmonics have recently been used in studies of morphogenesis in zebrafish and mouse (**Romeo et al., 2021**; **Dalmasso et al., 2021**).

Spherical harmonic basis functions are indexed by two parameters $(l, m)$, such that $l \geq 0, |m| \leq l$ representing respectively the degree and order of the harmonic (**Figure 3a**). A signal defined on the sphere can be written as a linear combination of such functions. Decomposing a signal into spherical harmonics hence amounts to finding the coefficients $f_{lm}$ of this weighted sum. In the case of

our spatiotemporal scalar strain rate field, the coefficients $f_{lm}$ are also a function of time and can be obtained as shown in *Equation 10*.

$$f_{lm}(t) = \oint f(\theta, \phi, t) Y_{lm}^*(\theta, \phi) dA \qquad (2)$$

Here, $Y_{lm}^*$ stands for the complex conjugate of the spherical harmonic $Y_{lm}$. Moreover, for a given degree $l$, each of the $(2l + 1)$ spherical harmonics $(Y_{lm})_{|m|<l}$ spatially partitions the unit sphere into as many spatial domains, indicating when a signal is positive, negative, or null (*Appendix 1—figure 4a*). *Figure 4a* illustrates the projections of the scalar strain rate field to spherical harmonics $(Y_{lm})_{l\leq2,|m|<l}$ at $t = 4.74$ hpf, and their mapping unto the surface of the embryo. These plots reveal, for instance, that while there is no embryo-wide-dominant morphogenesis process at this time ($l = 0, m = 0$), smaller regions, notably the vegetal pole, are experiencing significant morphological activity ($l = 1, m = 0$).

The contributions of each spherical harmonic to the global signal can be assessed more rigorously and interpreted in the light of biology. To this effect, we observe the temporal dynamics of the coefficients $f_{lm}(t)$ associated with each spherical harmonic. In analogy to *principal components analysis*, we measure the average variance ratio ($vr(Y_{lm}(t)) = |Y_{lm}(t)|/\sum_{l<L_{max},|m|\leq l}|Y_{lm}(t)|$) over time of each harmonic with respect to the original signal (*Figure 4b*). With an average variance ratio of 64.4%, the spherical harmonic $Y_{00}$, capturing embryo-wide morphological activity, contributes the most to ascidian morphogenesis. Spherical harmonic $Y_{10}$ is the next contributor, coming second with a variance ratio of 4.1%. This observation is warranted as $Y_{10}$ maps to the animal and vegetal poles of the embryo, which are the epicenters of synchronized cellular divisions and endoderm invagination, respectively (*Jeffery, 1992*; *Lemaire, 2009*). Interestingly, variances in the directions of $Y_{00}$ and $Y_{10}$ evolve in an antiphased pattern, most notably in earlier parts of the plot, with $Y_{00}$ contributing maximally (and $Y_{10}$ minimally), during periods of cell division, before relinquishing some variance shares to $Y_{10}$, which then peaks. This suggests that while sporadic deformations induced by cellular divisions often dominate the landscape of morphological activity, an observation consistent with studies in other species (*Cislo et al., 2023*), other localized, slower processes are at play in the embryo. The described pattern tends to fade out in the later parts of the plot, suggesting a shift in development dynamics.

Furthermore, by observing the time dynamics of the coefficients themselves (*Figure 4c*), one can easily identify which parts of the embryo are concerned by the morphological changes depicted. For instance, the positive peaks in $f_{10}(t)$ (*Figure 4c*) indicate that the morphological processes at hand take place in the northern hemisphere of the sphere. Remarkably, these coincide with rapid growth in cell population and thus synchronous cell divisions, which are known to be restricted to the animal pole of the ascidian embryo during endoderm invagination (*Jeffery, 1992*). In addition, most of the dynamics captured by $f_{10}(t)$ are in the negative spectrum ($f_{10}(t) < 0$), pointing to the lower hemisphere of the embryo, the foyer of several cell shape deformations at play in ascidian early development.

The sporadic short-time scale cell division events in the animal pole coexist with numerous other features of morphogenesis, most notably, the larger-scale continuous deformation process in endoderm invagination at the embryonic vegetal pole. Beside the peaks on the plots of the time series, it is not a trivial task to identify what other rich insights may be hidden in this data. A simple observation of the oscillatory patterns of these main modes hence paints an incomplete picture of ascidian morphogenesis. Extracting the footprint of all morphogenesis processes in these time series requires further analysis.

## Spectral decomposition in time: Wavelets analysis of spherical harmonic signals unveils the blueprint of morphogenesis

Analyzing time series often implies the understanding of how a signal is composed and how its components overlap in time. Wavelets have been put forward as effective multiresolution tools able to strike the right balance between resolution in time and resolution in frequency (*Torrence and Compo, 1998*; *Lau and Weng, 1995*). Although they have been taken advantage of in the broader context of biology, most notably in the analysis of brain and heart signals (*Brunton and Kutz, 2022*), they have so far been underused in developmental biology. The reason might be found in the reality that morphogenesis data is often not understood in terms of time series. Our spherical harmonics decomposition of morphogenesis, inspired by similar endeavors in other fields (*Dahlen and Tromp, 2021*; *Knaack and Stenflo, 2005*), offers an unprecedented opportunity to leverage the existing rich signal

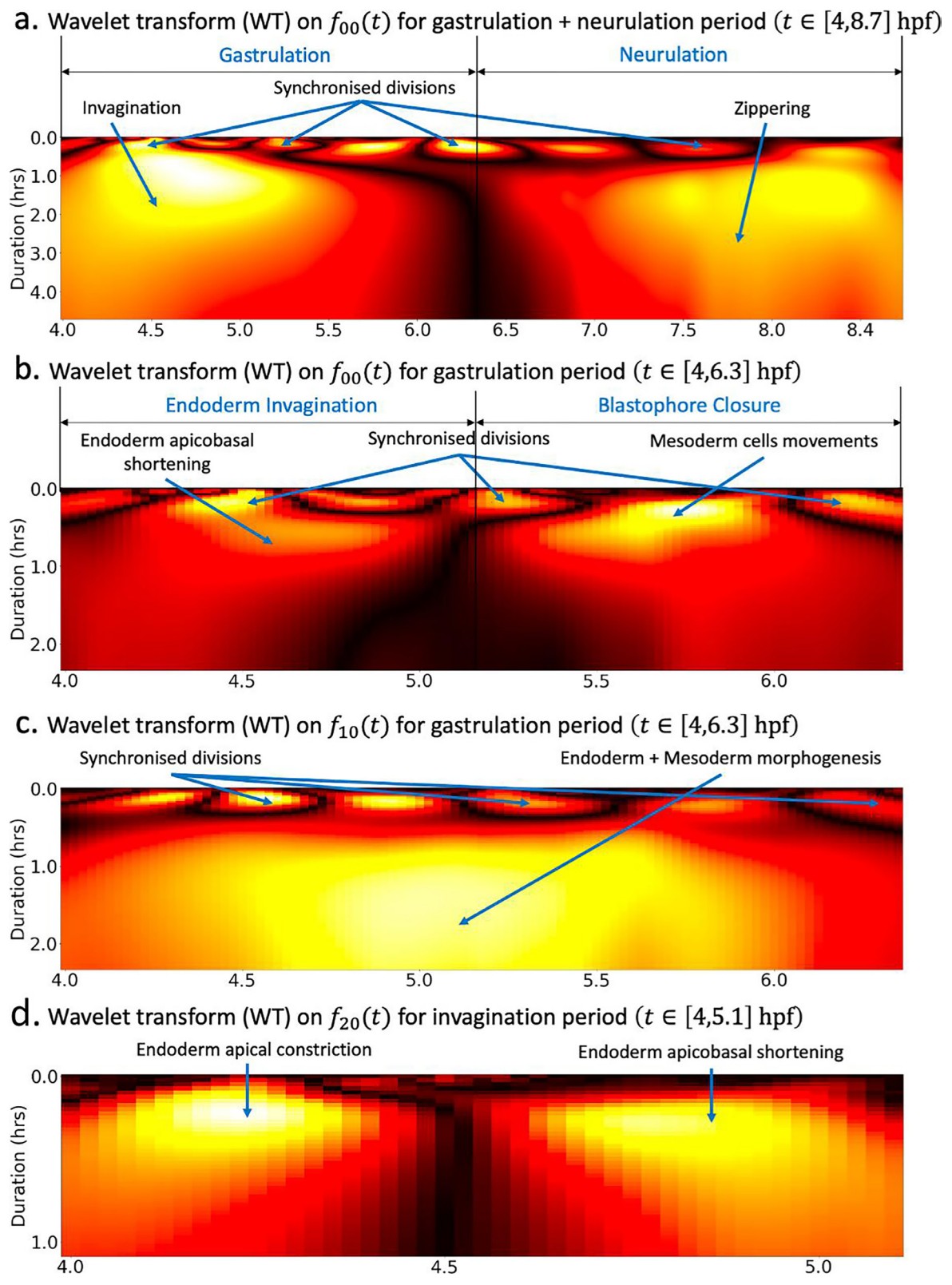

**Figure 4.** Wavelet analysis highlights multi-timescale modes of morphogenesis. (**a**) Scalogram resulting from the Ricker wavelet transform applied to $f_{00}(t)$ over the whole period covered by the dataset $t \in [4, 8.6]$ hpf. (**b**) Scalogram resulting from the Ricker wavelet transform applied to $f_{00}(t)$ restricted to the gastrulation period $t \in [4, 6.3]$ hpf. The high-frequency events highlighted here represent time points of synchronized division across the embryo. The dark band in the middle separating two large red regions indicates that there are two phases of invagination characterized by large

*Figure 4 continued on next page*

*Figure 4 continued*

deformations and a relatively calm transition phase in between. (**c**) Scalogram resulting from the Ricker wavelet transform applied to $f_{10}(t)$ restricted to the gastrulation period $t \in [4, 6.3]$ hpf. Similar to (**b**), the high-frequency events indicate synchronized division in the embryo. (**d**) Scalogram resulting from the Ricker wavelet transform applied to $f_{20}(t)$ restricted to endoderm invagination $t \in [4, 5.1]$ hpf.

processing toolbox in development biology. In particular, enlisting the help of wavelet transforms in unlocking the complex entanglements of the multiple morphological process at play during ascidian early development. We proceed to apply the Ricker wavelet transform to our spherical harmonics time series, normalized by mean and standard deviation in different time windows of interest. The result is a set of scalograms that decompose the signals into canonical components organized in timelines that reveal the story of ascidian morphogenesis.

First, we apply the wavelet transform on the time series $f_{00}(t)$ to the entire time range covered by our dataset, comprising both gastrulation and neurulation (*Figure 4a*). Mirroring this time series, the high-frequency events depicted by yellow blobs at the top of the heatmap represent periods of synchronized division across the embryo. The dark band in the middle separating two large regions depicts a short transition phase delimiting two phases of ascidian early development. The timing of these stages as reflected in the scalogram matches the timeline of gastrulation and neurulation. Within both phases, the concentric gradients from red to yellow culminating in dense yellow spots in the center of both regions portray increasing morphodynamics.

To better understand the specifics of ascidian gastrulation, we restricted the wavelet transform to the gastrulation period ($t \in [4, 6.3]$ hpf). The resulting scalogram (*Figure 4b*) shows that ascidian gastrulation unfolds itself in two major phases, delineated on the scalogram by the dark region at the center of the heatmap. The timeline of these events, strengthened by an analysis of topological holes in the embryo (*Figure 4b*), supports the hypothesis that these phases correspond to endoderm invagination followed by the near-closing of the future gut, a process initiated by the collective motion of lateral mesoderm cells known as *blastophore closure*. Both the time series (mostly in the negative spectrum) and the scalogram of $f_{10}(t)$ (*Figure 4c*) add another layer of validity to this conclusion: the large yellow blob occupying the majority the plot surface highlights that fact that regions of the embryo covered by spherical harmonic $Y_{10}$, hence endoderm and mesoderm cells are subject to intense and prolonged morphological processes.

The first of these two phases, namely endoderm invagination, has been thoroughly investigated in the literature. Most notably, it was identified that endoderm invagination was driven by two distinct mechanisms of endoderm single cells (*Sherrard et al., 2010*): first, cells constricted apically by reducing the surface area of the apices, flattening the convex vegetal pole of the embryo setting the stage for invagination. This was followed by animal vegetal shortening of their lateral faces, triggering endoderm invagination. The wavelet transform restricted to the period of endoderm invagination applied to $f_{20}(t)$, whose corresponding spherical harmonic function $Y_{20}(t)$ maps more precisely to the endoderm, beautifully captures this two-steps process (*Figure 4d*). The timing revealed by this scalogram is in accordance with an analysis endoderm cell shape ratios (*Figure 4b*).

## Spectral decomposition of morphogenesis in experimentally perturbed embryo

To assess how our framework adapts to different phenotypes, we set out to conduct a spectral decomposition of morphogenesis in an experimental manipulated embryo. In this particular mutant, MEK kinase was inhibited, which resulted in a massive re-specification of vegetal cell fates, and a disruption of endoderm invagination (*Guignard, 2020*). We applied to the mutant dataset (*Figure 5a*, top) each of the steps in our workflow. First an evolving mesh matching the shape of the embryo at every time point was obtained through the level set scheme. Then, a strain rate tensor field was computed over the surface of the embryo throughout development time (*Figure 5a*, bottom). A spatiotemporal spectral analysis was subsequently conducted using spherical harmonics on the mutant surface and wavelet analysis of the time series of the coefficients of the main harmonic modes. In order to meaningfully compare the dynamics of the mutant development against those of the wild-type embryo, the analysis was carried out at the 64-cell stage.

Similar to the WT embryo, the main harmonic modes in the mutant development were $Y_{00}$ and $Y_{10}$ with respective variance ratios 73.68% and 1.65%, making the time series $f_{00}(t)$ and $f_{10}(t)$ the main

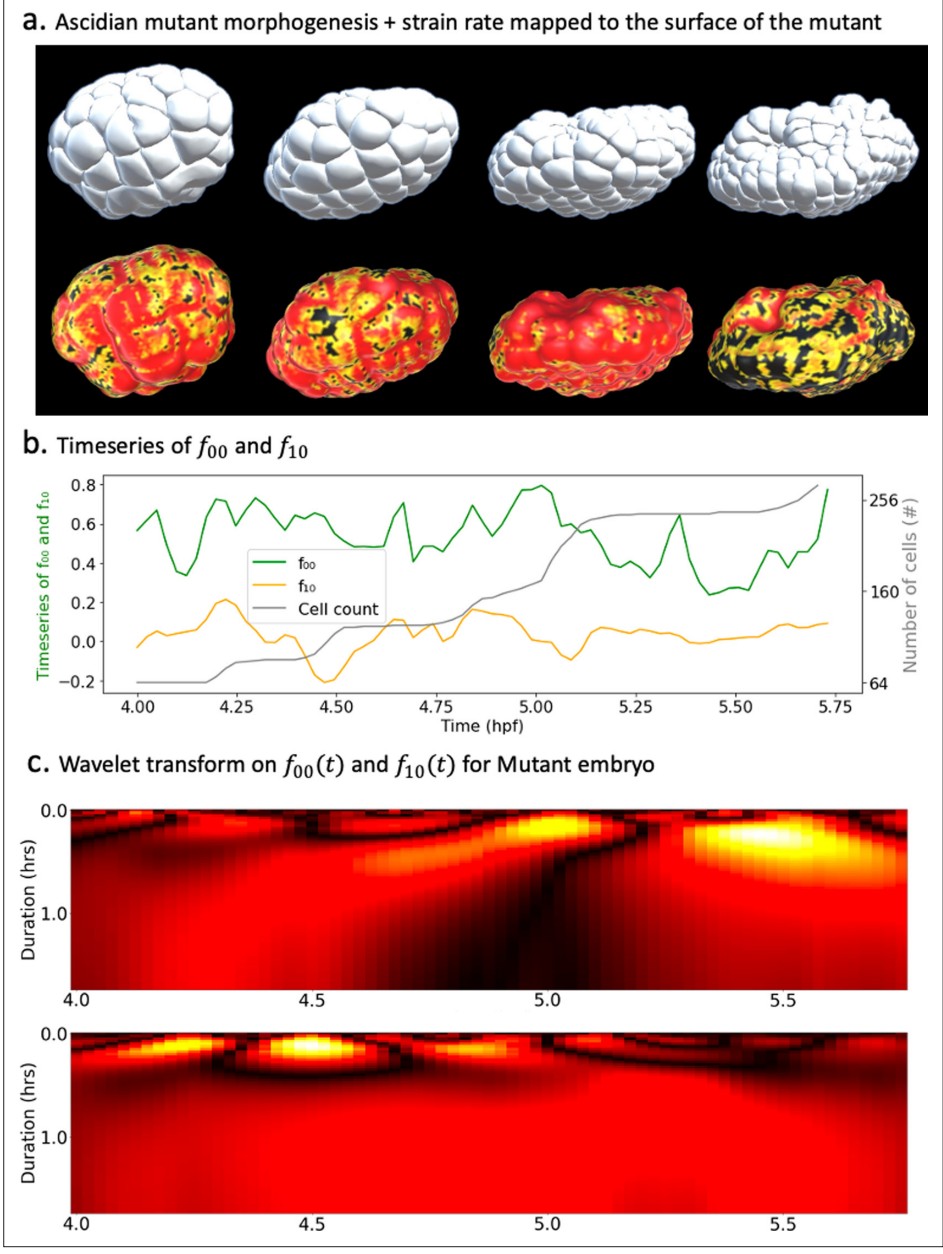

**Figure 5.** Spectral decomposition of morphogenesis in mutant embryo. (**a**) Top: ascidian mutant morphogenesis. Bottom: spatiotemporal scalar strain rate field mapped to the mutant surface. (**b**) Time evolution of the coefficients $f_{00}$ and $f_{10}$ associated with spherical harmonics ($Y_{00}$ and $Y_{10}$). The cell population dynamic is also included in the plot for clarity. (**c**) Wavelet transform applied on $f_{00}(t)$ (top) and $f_{10}(t)$ (bottom).

focus of our examination. The temporal dynamics of these coefficients already reveal major differences between the two strains of ascidians (**Figure 5b**). On the one hand, the drop in the share of $Y_{10}$ is telling of the lower order of morphological activity in the vegetal hemisphere. The difference with WT embryos is even more striking when considering that they are deprived of cell division in their vegetal hemisphere. On the other hand, the peaks and lows of $f_{10}(t)$ that coincide with growth in cell numbers are not restricted either to the negative or positive domain of the curve. This implies that, contrary to the WT, synchronous cell divisions are not restricted to one hemisphere of the embryo.

The wavelet transform applied to time series $f_{00}(t)$ and $f_{10}(t)$ yield scalograms, which bring further insights into the mutant morphogenesis (**Figure 5c**). On the one hand, the heatmap of $f_{00}(t)$ hints to two distinct phases of the mutant development during the observed time span (**Figure 5c**, top).

The timing of the second phase seems to coincide with the advent of the major wave of cell divisions in the embryo. On the other hand, plots of the scalogram of $f_{10}(t)$ appear to concur with the hypothesis of very low morphological activity in the vegetal pole. The constant red on this heatmap reflects unchanging levels of morphological activity at the vegetal pole of the embryo and confirms the absence of drastic cell deformations that usually drive invagination. This is in alignment with the perturbation induced by MEK kinase that prevents invagination from happening in the mutant endoderm.

## Discussion

Ray Keller's roadmap of morphogenesis studies establishes a clear path for understanding the biomechanical processes involved in development (*Keller et al., 2003*). In his proposed workflow, the first step is to determine when and where cells move. Identifying regions of significant morphological activity in space and time has usually followed a script consisting of observing via a microscope the developing system, formulating a hypothesis of what is happening in the system, and subsequently affirming or refuting the hypothesis using qualitative analysis. This method, which has successfully propelled the field of developmental biology to its current heights, nevertheless has some limitations. Distinct morphogenesis events can overlap both in space and time, rendering eye observation vulnerable to misinterpretation. Second, these methods are not automated, hence do not scale.

In this work, we attempted to develop an alternative approach to probing development in living systems. Our approach takes advantage of the recent boom in the availability of single-cell shape tracking data to propose a generic method able to identify interesting defining morphological processes through space and time in developing embryos. The method takes as input data consisting of evolving cell geometries and outputs a series of spatial heatmaps showcasing in the time–frequency domains the most salient traits of morphogenesis in the studied embryo. There is, however, no requirement for segmented cells: the method can be extended to accommodate microscopy imaging data, a feature available in the code submitted. Our framework presents over the traditional eye test method multiple advantages. First, the workflow is fully automated, providing an unprecedented hands-off approach in preliminary studies of morphogenesis. Another outstanding advantage of our workflow over traditional methods is that our workflow is able to compress the story of the development, such that, in a single image, one can grasp the essence of morphogenesis in a system of interest. In particular, our method has been able to neatly discriminate between the gastrulation and neurulation phases of ascidian early development, identify a second phase of gastrulation, *blastophore closure*, which follows invagination, reconstitute the two-step process of endoderm invagination during the gastrulation phase, while clearly distinguishing between short-scale division events and low-frequency tissue-wide deformations.

In order to achieve this fit, raw cell shape data underwent a series of transformations, including a level sets-driven homeomorphic map of the unit sphere to the developing embryo's surface, the computation of the strain rate field of embryo deformations through time using successive iterations of this map, a spherical harmonics decomposition of this strain rate field, and wavelet decomposition of the most significant spherical harmonics time series. Each of these transformations comes with its own challenges, but also delivers new perspectives for the study of living systems. Our level set scheme excels at defining a homeomorphic map between the unit sphere and surface of the embryo. It goes without saying that in order for the deformed sphere to best match the shape of the embryo, a high sampling of points on the unit sphere is required. A compromise is, however, necessary between this sampling and, on the one hand, the overall spatial resolution of the original dataset, on the other hand, the induced computational complexity. In its current form, the scheme produces approximations of Lagrangian particles only under the assumption of small deformations in the embryo. Hence, the sampling rate during microscopy imaging is of critical importance: the shorter intervals between two successive frames of the movie, the more Lagrangian-like the particles are expected to behave.

Given the provision of tracked surface particles meshed at every frame in a triangular network, the evaluation of the strain rate field is straightforward and enables, among others, a unified description of complex cell-level and tissue-level dynamics (*Blanchard et al., 2009*), such as drastic deformations and synchronized divisions. The accuracy of this field is affected, as previously, by both the spatial sampling of material points on the unit sphere and the timely sampling of morphogenesis frames. Despite the richness in its tensor form, a visualization of the eigenvalue field derived from

this tensor field on the surface of the embryo can already highlight significant processes in morphogenesis. The decomposition of this field into spherical harmonics allows a better appreciation of the spatial patterns of morphological activity in the embryo, each harmonic mapping a region of space. Our spherical harmonics decomposition of morphogenesis results in a set of time series of coefficients associated with each harmonic, representing, to the best of our acknowledge, the first comprehensive time series-based description of morphogenesis. This transformation unleashes the full power of signal processing tools into studies of morphogenesis. The basis of spherical harmonics being infinite, a challenge here is to discriminate between harmonics that significantly contribute to the composed signal and those that do not. Here, this was done by singling out harmonics that contributed the most to the variance of the strain rate field. Furthermore, the filtration of principal harmonics modes enables the representation of morphogenesis in a significantly compressed form in comparison to the initial datasets. This lower-dimensional representation of morphogenesis can be helpful, among others, in modeling the physical dynamics of the system (*Romeo et al., 2021*). Another challenge is with the interpretability of the harmonics, which is subject to the alignment of the embryos. The datasets used in this article presented the advantage that their *y*-axis was quasi-aligned with their *vegetal − animal* axis. For embryos that do not have this property, prior processing to align them will be required. Alternatively, rotation-invariant representations can be used to appropriately interpret the harmonics.

Despite describing canonical interactions in the space of spherical harmonic functions, our spherical time series still represent composed signals in time. We use the *Ricker* wavelet as a mathematical microscope to zoom-in and zoom-out through these signals in order to identify their fundamentals components. This operation results in two-dimensional time–frequency heatmaps that showcase, for each time series, the footprint of its canonical high- and low-frequency components, which can be mapped to biological processes. The sum of these tell the story of morphogenesis in the region corresponding to the spherical harmonic. Here, it might also be useful to wisely target windows of time of interest and normalize the data such that interesting transitions can be picked up easily by the wavelet transforms. The resulting heatmaps can be fed to analytic workflows such as deep neural networks for further studies. Example scenarios could include variational studies of morphogenesis processes in different WT or mutant embryo. Furthermore, the workflow presented in this article can be applied to the examination of single-cell morphological behaviors in development.

## Acknowledgements

This research benefited from funding from the NSF-Simons Center for Quantitative Biology at Northwestern University (NSF: 1764421 and Simons Foundation/SFARI 597491-RWC), the Physics Frontier Center for Living Systems funded by the National Science Foundation (PHY-2317138), and Google Cloud for Research.

## Additional information

### Funding

| Funder | Grant reference number | Author |
|---|---|---|
| National Science Foundation | 1764421 | Joel Dokmegang Madhav Mani |
| National Science Foundation | PHY- 2317138 | Edwin Munro |
| Simons Foundation | 597491-RWC | Joel Dokmegang Madhav Mani |

The funders had no role in study design, data collection and interpretation, or the decision to submit the work for publication.

### Author contributions

Joel Dokmegang, Conceptualization, Resources, Data curation, Software, Formal analysis, Validation, Investigation, Visualization, Methodology, Writing – original draft, Project administration, Writing

– review and editing; Emmanuel Faure, Resources, Data curation, Software, Supervision, Investigation, Methodology, Project administration; Patrick Lemaire, Data curation, Supervision, Validation, Methodology, Writing – review and editing; Edwin Munro, Conceptualization, Supervision, Funding acquisition, Validation, Investigation, Methodology, Writing – review and editing; Madhav Mani, Conceptualization, Formal analysis, Supervision, Funding acquisition, Validation, Investigation, Methodology, Writing – original draft, Project administration, Writing – review and editing

## Author ORCIDs
Joel Dokmegang (ID) https://orcid.org/0000-0001-9953-7913
Emmanuel Faure (ID) https://orcid.org/0000-0003-2787-0885
Patrick Lemaire (ID) https://orcid.org/0000-0003-4925-2009
Madhav Mani (ID) https://orcid.org/0000-0002-5812-4167

Reviewer #1 (Public review): https://doi.org/10.7554/eLife.94391.3.sa1
Reviewer #2 (Public review): https://doi.org/10.7554/eLife.94391.3.sa2
Author response https://doi.org/10.7554/eLife.94391.3.sa3

## Additional files

### Supplementary files
MDAR checklist

### Data availability
The raw imaging data used in this study has been supplied by the Lemaire and Faure groups through the Morphonet platform (https://morphonet.org/; *Leggio et al., 2019*). The code written for the analyses in this study is available at https://github.com/guijoe/lmg, copy archived at *Dokmegang, 2025*.

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

# Appendix 1

## An overview of ascidian early development

Owing to the relative simplicity of their development, ascidians are an interesting model of development for biologists (*Lemaire, 2009*). On the one hand, ascidians embryos are optically clear, which makes them suitable for 3D imaging, observation, and the analysis of the dynamics of their cell shapes in fixed and live samples (*Lemaire, 2009*). On the other hand, ascidians display rapid development. Whereas the zygote to gastrula transition usually requires several days in other organisms, this process is remarkably fast in ascidian embryos, spanning only a few hours (*Appendix 1—figure 1a*). At the onset of gastrulation, a typical embryo is made of about a hundred cells spatially organized into the endoderm, mesoderm, and a cellular mass of unspecified fate (*Lemaire, 2009*). Gastrulation is the first large-scale morphogenetic process in development, involving drastic rearrangements of cells that reorganize to lay grounds for the future body plan. Lewis Wolpert's famous quote *"It is not birth, marriage or death, but gastrulation which is truly the most important time in your life"* (*Hopwood, 2022*) emphasizes the significance of this process in development. Similar to other large-scale processes (*Leptin, 2005*; *Godard and Heisenberg, 2019*), ascidian gastrulation is characterized by a dynamic interplay between proliferation and tissue mechanics.

The most dominant tissue-wide movement in ascidian gastrulation is the invagination of the endoderm lineage, which fosters the creation of a gut at the vegetal pole of the embryo. A particularity of invagination in this context is its signature two-step sequence (*Sherrard et al., 2010*; *Fiuza et al., 2020*). In the first step, single endoderm cells constrict their apical faces, triggering a coordinated movement that results in the flattening of the endoderm epithelia. In the second step, proper invagination occurs as a consequence of the contraction of endoderm cells apicobasal axes and the expansion of their basal faces (*Sherrard et al., 2010*; *Fiuza et al., 2020*). *Appendix 1—figure 1b* illustrates these findings by showing a sharp rise followed by an equally acute decline in the contribution of endoderm cells apicobasal axis to the variance in their shape. As they constrict their apical faces, the variance across secondary axes is reduced and transferred to the apicobasal. Furthermore, as they shorten their apicobasal axis, the accumulated variance is distributed across secondary axes. The combination of these two steps results in a relatively swift endoderm invagination (*Sherrard et al., 2010*), as shown by the rapid increase in gut radius (*Appendix 1—figure 1c*).

Proliferation also registers as a preponderant mechanism in ascidian development. Through multiple rounds of cellular divisions, the cell population increases approximately threefold during gastrulation (*Nishida, 1986*, *Lemaire, 2009*) and tenfold through neurulation (*Appendix 1—figure 1b*). The properties of cellular division during ascidian gastrulation depend on the embryonic hemisphere on which they are located. Divisions at the animal pole display more regularity than those occurring at the vegetal pole. On the one hand, they have been found to be mostly synchronized in time (*Nishida, 1986*). On the other hand, through successive rounds, cells alternate their division plane by choosing a plane perpendicular to the previous round (*Nishida, 1986*). In contrast, fewer divisions happen at the vegetal pole, and no consistency has been observed in the orientation of divisions (*Nishida, 1986*). The question of whether these mechanisms influence each other during ascidian gastrulation has also been examined. It was found that embryos in which cellular divisions had been stopped via treatment with noccodazole proceeded without issue to endoderm invagination (*Sherrard et al., 2010*), demonstrating that proliferation is not required for invagination to occur.

As in several species, the development program in ascidians features neurulation after gastrulation. This important stage consists of transforming the neural plate into the neural tube via the closure of the fold created during gastrulation. Following invagination, cells at the periphery of the gut initiate its closure by inducing a coordinated movement toward the center. It has been observed that neural tube closure in ascidians evolves in a way similar to the zippering of a hand bag, whereby the gut is closed by the gradual action of cells establishing connections along a linear trajectory (*Hashimoto et al., 2015*). The zippering is carried out by the coordinated interactions of about 80 cells (*Hashimoto et al., 2015*), making it a morphological stage with significant mechanical footprint in the storyline of ascidian development.

## Level set scheme

Here, we propose a method that consists of computing a unique polygonal mesh with fixed vertex networks that evolves to match the shape of the embryo at every single development frame. This method is adapted from *Zhao et al., 2000*. We consider the following energy, which quantifies the global distance between the two surfaces ($S_1(t)$: real embryonic surface) and ($S_2(t)$: simulated embryonic surface) (*Equation 3*). This energy reaches a global minimum when the global distance between ($S_1(t)$) and ($S_2(t)$) goes to zero. Therefore, with sufficiently enough vertices regularly distributed on ($S_2(t)$), both surfaces can be brought to share a similar geometry simply by allowing ($S_2(t)$) points to move along the path of descendant gradients on the energy landscape defined by equation *Equation 3*.

$$E(S_1(t)) = \left( \int_{S_2(t)} d^2(x, S_1(t)) ds \right)^{(1/2)} \tag{3}$$

Here, $d(x, S_1(t))$ is the distance between a point $x$ of $S_2(t)$ to $S_1(t)$ defined as the minimal distance $d(x, x')$ where $x'$ is a point of $S_1(t)$. $ds$ is the elementary surface area surrounding the vertex with position $x$ on ($S_2(t)$).

For a vertex on ($S_2(t)$) with position $x$, we have the following equation describing its trajectory in the transformation from ($S_2(t)$) to ($S_1(t)$) (*Equation 4*).

$$\frac{dx}{d\tau} = -\frac{\delta E}{\delta x} \tag{4}$$

*Zhao et al., 2000* show that

$$\frac{\delta E}{\delta x} \approx [\frac{d(x)}{d_{max}}][\nabla d(x) \cdot n + \frac{1}{2} d(x)\kappa]n \tag{5}$$

Here, $d_{max} = \max d(x) \approx \left( \int_{S_2^T} d^2(x, S_1^T) ds \right)^{(1/2)}$, $n$ and $\kappa$ are, respectively, the normal vector and the mean curvature on ($S_0^t$) at position $x$.

Feeding back Equation 5 into 4, we find

$$\frac{dx}{d\tau} = -[\frac{d(x)}{d_{max}}][\nabla d(x) \cdot n + \frac{1}{2} d(x)\kappa]n \tag{6}$$

Hence, we have the following update rule for a vertex with position $x_i$ in our discrete setting using an explicit Euler scheme.

$$x_i^{\tau+1} = - \left( \nabla_i d \cdot n_i + \frac{1}{2} d_i \kappa \right) n_i \times \Delta\tau + x_i^\tau \tag{7}$$

In theory, the scheme such defined is able to bridge the gap between a topological sphere and the embryonic shape. However, the practical implementation of this scheme presents a few challenges.

The first issue we face is that of a proper definition of the surface of the embryo ($S_1(t)$). We recall that our input data can take the form of a discrete set of volumetric cells spatially organized into an embryo, rather than a continuous surface. In this case, we construct the embryonic surface by extracting surface triangles from all external cells and endowing this set with a topology by defining triangle neighborhoods. For computational efficiency, neighborhoods are first computed between cells, and then between triangles of neighboring cells. From this construction, it follows that the distance of a point of space to the surface of the embryo is computed as the minimal distance from this point to all triangles of ($S_1(t)$). In the case that embryonic shape data is in a pixel format, a simple transform of (($S_1^+(t) - S_1(t)$))—where $S_1^+(t)$ is the dilation operation applied to $S_1(t)$—can be used to extract the embryonic surface.

The second challenge is the issue of the initialization of the homeomorphic surface on which the scheme will be applied. For the initial frame ($t = 0$) of the dataset, in the absence of any reference shape for ($S_1(0)$), ($S_2(0)$) is initialized to a sphere centered at the center of mass of the embryo and whose radius is equal to the maximal radius of the embryo, measured from its center of mass. For subsequent frames ($t$), in order to maximize computational efficiency and minimize compounding

errors arising from the explicit scheme, $(S_2(t))$ is initialized to a weighted average between the minimal sphere circumscribing $(S_1(t))$ and $(S_2^{'}t - 1))$. Other methods of initialization can be used, including a sparser version of $(S_1(t))$ on which vertices are added.

Inherent to this initialization problem is the question of the number of vertices required to meaningfully represent the embryonic surface. It is noteworthy here to recall that the surface of the embryo riddled with curved apical faces of cells and discontinuous cellular boundaries, which increase with time: the development time spanned by the dataset features cell population sizes ranging from 64 surface cells all the way up to 562 surface cells. The basic spherical shape is obtained from an icosahedron of 12 vertices whose distances to the center are normalized. Additional vertices are obtained by iterative *butterfly* subdivisions of this spherical mesh (*Hardy and Steeb, 2008*). *Appendix 1—figure 2* portrays how well simulated cell junctions (green points) mirror the behavior of actual cell junctions (red points) as a function of the number of mesh vertices chosen. As the number of vertices increases, the average relative distance between real and simulated material particles, normalized by the average length of cell apical faces, decreases (*Appendix 1—figure 2b*). However, the relative between simulated and real material particles at consecutive frames does not vary much with the number of vertices (between 5% and 8%; *Appendix 1—figure 2c*). The slight increase observed is most likely due to the need for longer iterations of the gradient descent scheme as the number of vertices grows.

The level set scheme thus described is applied to every frame of development, transforming the initial dataset from a time series of cellular meshes with arbitrary topologies to a single surface mesh with unique topology, but evolving shape, matching the surface of the embryo at each frame.

## Strain rate computation

The strain rate tensor field is computed as the symmetric part of the gradient tensor of the velocity field. In order to minimize undesired local artifacts arising, for example, from the inherent errors to experimental measurements and the determination of the embryo surface mesh, the field is smoothed using a Gaussian mask spanning the one-ring and two-ring neighborhood of each vertex. Analytically, the determination of the strain rate tensor field obeys the equations in (8):

$$D(t) = \frac{1}{2}\left(L(t) + L^T(t)\right) \tag{8}$$

Here, $L(t) = \nabla v(t)$ and $v(t) = \frac{1}{\Delta t}\left(x(t + \Delta t) - x(t)\right)$, and $x(t)$ is the position of the particle at time $t$.

The determination of the gradient of the velocity field on the mesh entails the determination of different partial derivatives with respect to all spatial dimensions. This is not straightforward in our case of a discrete closed surface embedded in the 3D space. We therefore rely on the work of *Mancinelli et al., 2018* on the estimation of gradient fields on triangular meshes to derive expressions for these terms.

In their work, the gradient $\nabla_i f$ of a field $f$ at given vertex $i$ is computed as the mean value over all adjacent triangles $\mathcal{T}(i)$ of the restriction to each adjacent triangle of the field gradient $\nabla_i f_{ijk}$ at $i$. The sum is weighted by the area of each triangle $S_{ijk}$.

$$\nabla_i f = \frac{1}{S_i} \sum_{j,k|ijk \in \mathcal{T}(i)} \nabla_i f_{ijk} S_{ijk} \tag{9}$$

On each adjacent triangle, the field is assumed to be linear and the gradient is evaluated as follows:

$$\nabla_i f_{ijk} = \frac{1}{2S_{ijk}}[(f_j - f_i)n_{ik} + (f_k - f_i)n_{ij}] \tag{10}$$

Here, $n_{ij}$ and $n_{ik}$ represent the normal vectors to edges $ij$ and $ik$, respectively.

## Spherical harmonics

Spherical harmonics represent a basis of harmonic functions that are useful to decompose signals defined on a sphere. Each of the spherical harmonic functions partitions the sphere into regions of space where the signal is positive, negative, or equal to zero (*Appendix 1—figure 4*). Complex spherical harmonic functions are analytically defined as follows:

$$Y_{lm}(\theta, \phi) = \sqrt{(2l+1)\frac{(l-m)!}{(l+m)!}} P_{lm}(\cos\theta)e^{im\phi} \tag{11}$$

Here, $P_{lm}(\mu) = (1 - \mu^2)^{m/2}\frac{d^m}{d\mu^m}P_l(\mu)$ and $P_l(\mu) = \frac{1}{2^l l!}\frac{d^l}{d\mu^l}(\mu^2 - 1)^l$

A function $f$ defined on the unit sphere can be expanded using the spherical harmonic basis as follows:

$$f(\theta, \phi) = \sum_{l=0}^{+\infty} \sum_{m=-l}^{l} f_{lm}Y_{lm}(\theta, \phi) \tag{12}$$

Here, the coefficients $f_{lm}$ are obtained as $f_{lm} = \oint f(\theta, \phi)Y_{lm}^*(\theta, \phi)dA$.

From these coefficients, an approximation of the original signal can be constructed. Although the spherical harmonics form an infinite basis, in practice, we are restricted to expanding a function up to a maximum degree of $L_{max}$. *Appendix 1—figure 4* shows the decomposition and reconstruction of the strain rate scalar field signal using spherical harmonics. It can be visually observed that as $L_{max}$ increases, the reconstructed signal tends to be closer to the scalar strain rate field (*Appendix 1—figure 4c*), hence gradually capturing finer patterns of the scalar strain rate field. In contrast, but none less interesting, lower values of $L_{max}$ tend to better highlight larger spatial patterns of the strain rate field tensor, suitable for the identification principal modes of the studied dynamics, and regions of significant morphological activity (*Appendix 1—figure 4c*).

## Wavelet analysis: The Ricker wavelet

Like the Fourier transform, wavelets rely on parametrized kernel functions to represent a given signal. These kernel functions are parametrized in such a way that they can be stretched, shrunk, and shifted over the domain spanned by the time series to adapt to flexible windows in the time–frequency domain. These properties allow wavelets to pick up short-lived effects of high-frequency processes in the signal in narrow bands of time while also capturing background effects of low-frequency processes. Wavelets can, however, not indefinitely zoom-in or zoom-out through a signal domain as they are constrained by the so-called 'uncertainty principle'. The multiresolution of the wavelet transform is hence achieved at the price of a compromise between precision in time and resolution in frequency.

Multiple wavelets kernel exist and are fit for different purposes. In this work, we have made use of the Ricker wavelet to analyze the time series of spherical harmonic coefficients. The Ricker wavelet can be defined as the negative normalized second derivative of a Gaussian function. Its analytical expression is given as follows:

$$\psi(x) = \frac{2}{\sqrt{3\sigma}\pi^{1/4}}\left(1 - \frac{x^2}{\sigma^2}\right)e^{-\frac{x^2}{2\sigma^2}} \tag{13}$$

*Appendix 1—figure 6* shows an illustration of the Ricker wavelet transform applied on a composed signal $y(t) = sin(t) + sin(4t)$. Like the Fourier transform, the wavelet can distinguish between the high-frequency component ($sin(t)$, tiny yellow blobs at the top of the scalogram) and the low-frequency component ($sin(4t)$, large yellow blobs) of the signal. In addition, the wavelet also indicates when the frequencies occur in the signal (*x*-axis), and for how long they last (*y*-axis), thus providing high resolution in both the time and frequency domains.

## Endoderm dynamics: Cell aspect ratios and topological holes

To firmly ground the findings of our spectral decomposition of morphogenesis, we perform a more targeted analysis of the temporal patterns of endoderm morphogenesis. We observe two characteristics of the length of the apical-basal axis of endoderm cells, and the radius of the gut created by endoderm invagination. *Appendix 1—figure 7* presents results that perfectly align with the spectral analysis.

In *Appendix 1—figure 7a*, the blue plot represents the average variance of endoderm cells material particle positions across the apicobasal axis during gastrulation. The variations between 4.25 hpf and 5.0 hpf show the two steps of invagination. First, the variance across the AP axis increases rapidly as a result of a decrease in apical surface area. Then, this quantity plummets as a result

of apicobasal shortening. The red plot describes the temporal dynamics of endoderm gut radius created by endoderm invagination. The radius peaks around 5.2 hpf, which marks the transition between the invagination and the blastophore closure phases of ascidian gastrulation. This radius is computed at every time point using persistent homology, a branch of topological data analysis, which yields itself suitably to the study of topological features such as connected components, voids, and holes in point clouds (*Carlsson, 2009*).

Over the entire point cloud consisting of cellular mesh vertices, we sample points from a thin slice of space encompassing material particles of endoderm cells (*Appendix 1—figure 7a*). The resulting point cloud is then probed for holes. Before invagination, this region is dense with cellular material particles. However, as invagination progresses, particles leave, creating holes of greater radii between them, which can be captured by persistent homology. The red plot in *Appendix 1—figure 7a* corresponds to the radius of the maximal hole at each time.

**a.** Different views of a developing ascidian embryo

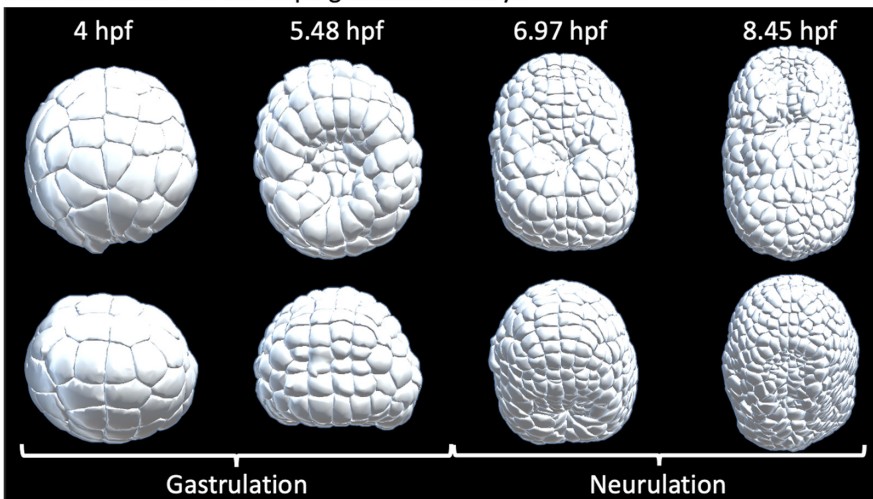

**b.** Cell population growth

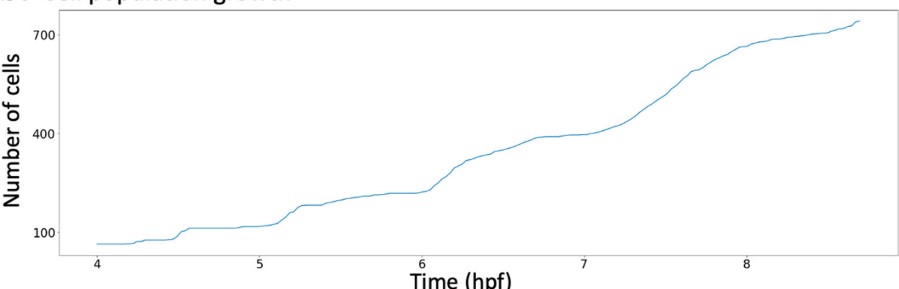

**Appendix 1—figure 1.** Ascidian early development gastrulation. Tthe early ascidian embryo goes through the phases of gastrulation and neurulation. (**a**) Timelines of gastrulation and neurulation in a developing ascidian embryo. (**b**) Cell population dynamics.

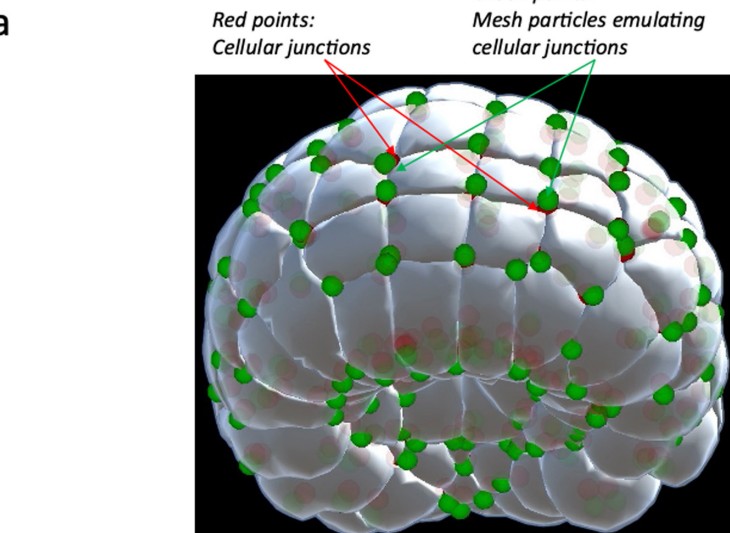

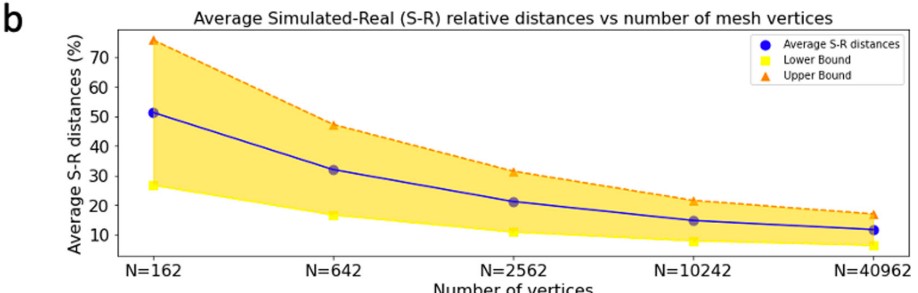

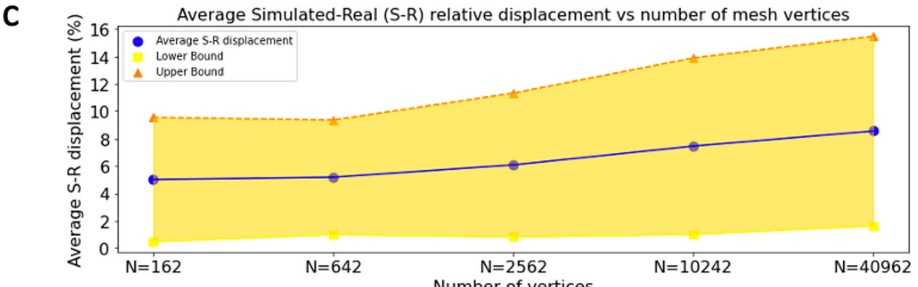

**Appendix 1—figure 2.** Level set scheme. (**a**) Real cell junctions (red) and simulated cell junctions (green) mapped unto the embryo. (**b**) Average relative distances between real and simulated cell junctions as a function of number of mesh vertices. The distances are normalized by the average side length of cell apices. (**c**) Average relative displacement between real and simulated cell junctions as a function of number of mesh vertices.

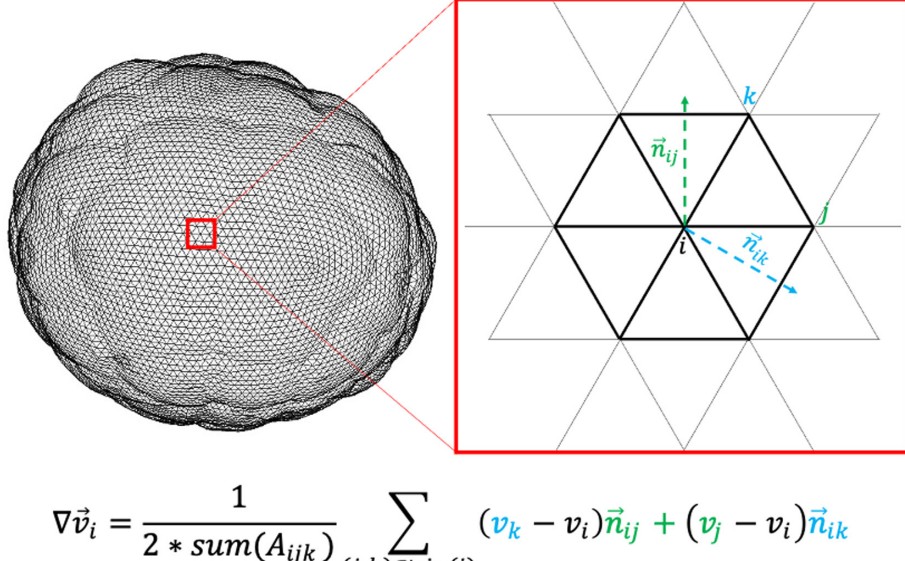

$$\nabla \vec{v}_i = \frac{1}{2 * sum(A_{ijk})} \sum_{(j,k)\in \text{tris }(i)} (v_k - v_i)\vec{n}_{ij} + (v_j - v_i)\vec{n}_{ik}$$

**Appendix 1—figure 3.** Discrete gradient computation.

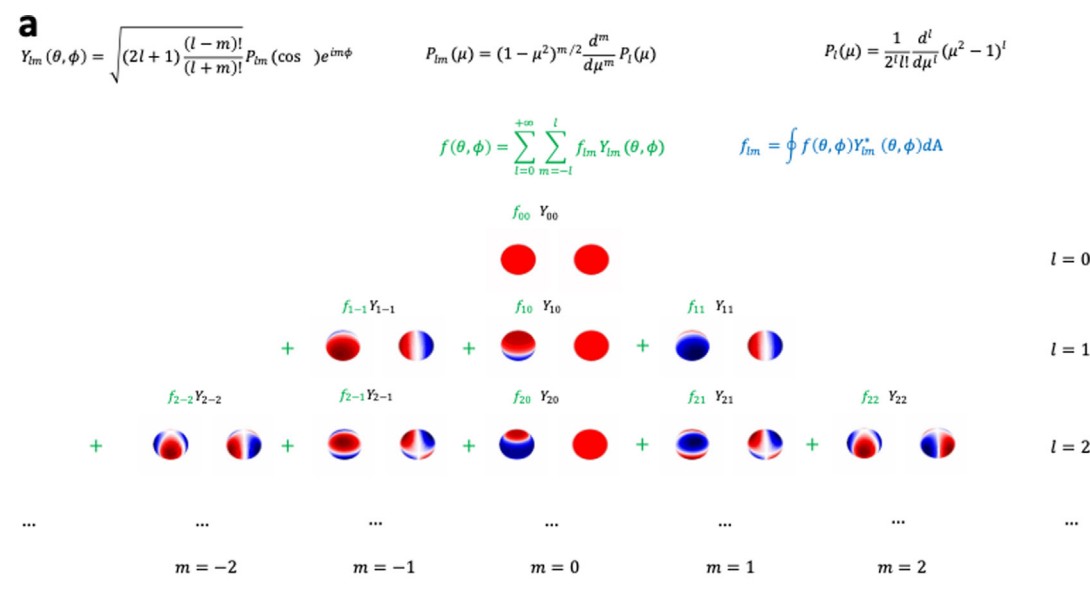

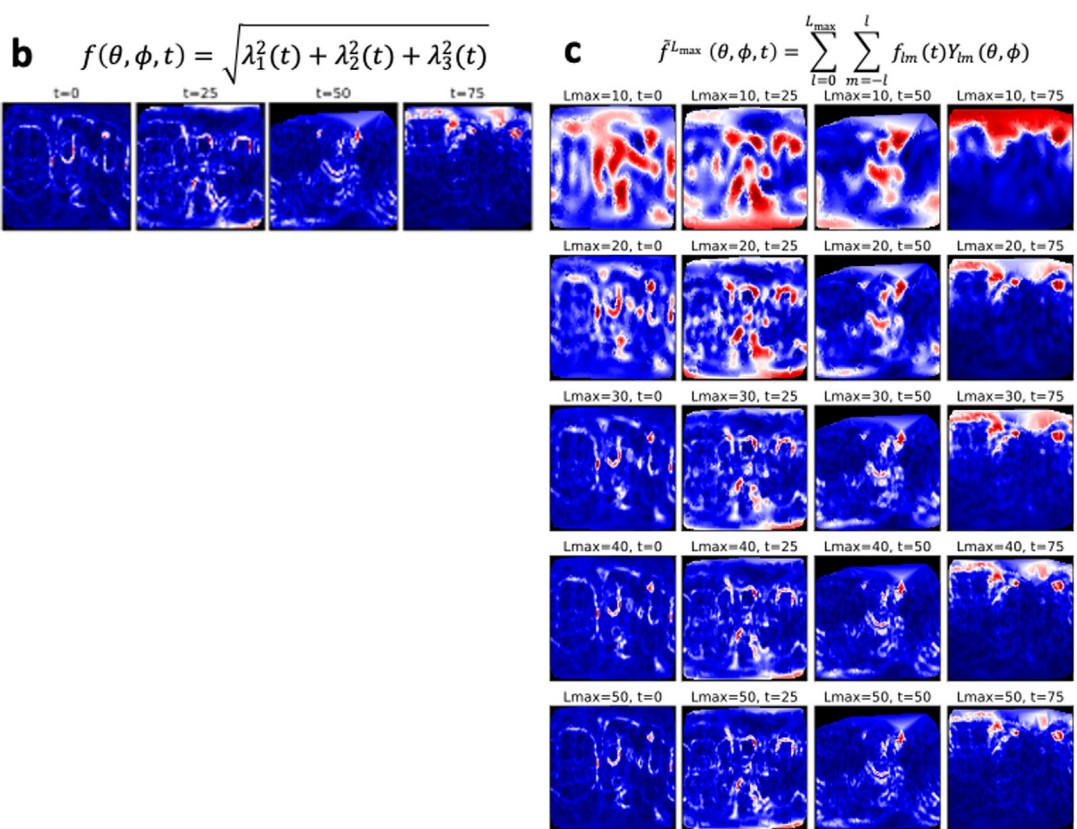

**Appendix 1—figure 4.** Spherical harmonics. (**a**) Analytical expression and spatial representation of spherical harmonics. (**b**) 2D representation of the scalar strain rate field. (**c**) Reconstructed scalar strain rate field based on the spherical harmonics decomposition up to a degree $L_{max}$.

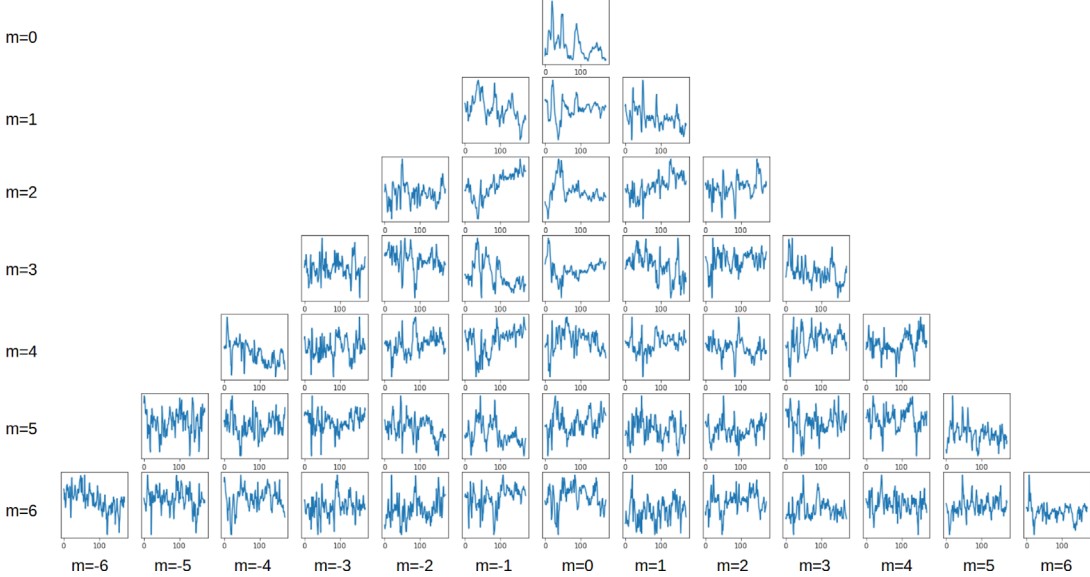

**Appendix 1—figure 5.** Spherical harmonics decomposition. Time series of the spherical harmonics coefficients up to l=6.

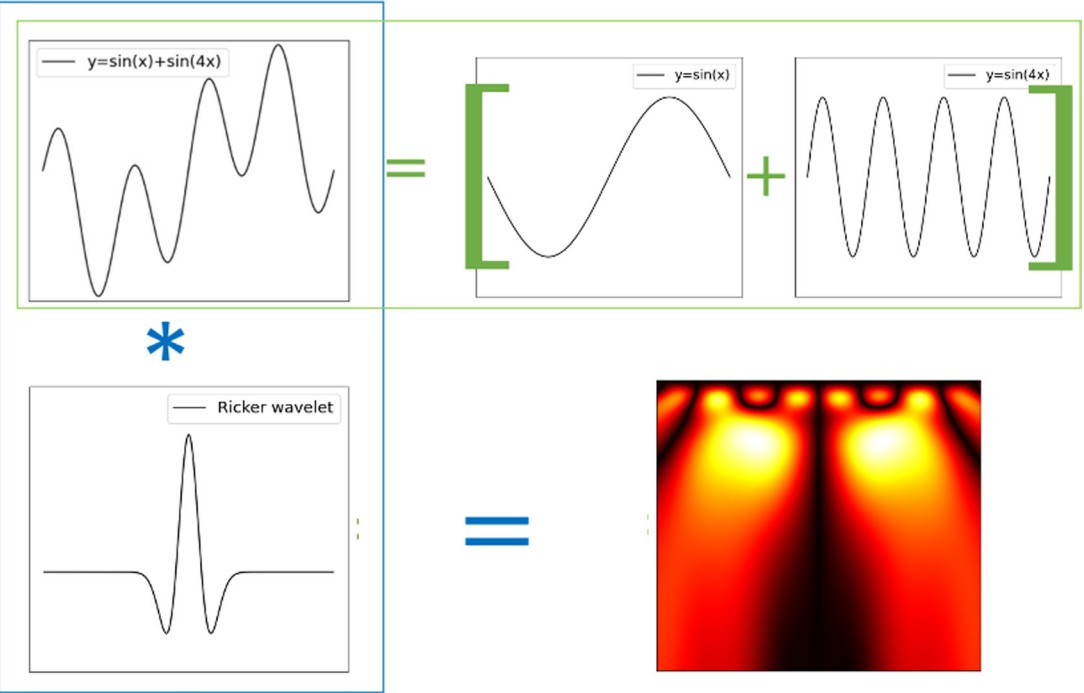

**Appendix 1—figure 6.** Wavelet transform. Ricker wavelet transform of a composed signal $sin(t) + sin(4t)$. The transform decomposes the signal into its canonical constituents: small yellow blobs for $sin(t)$ and large yellow blobs for $sin(4t)$.

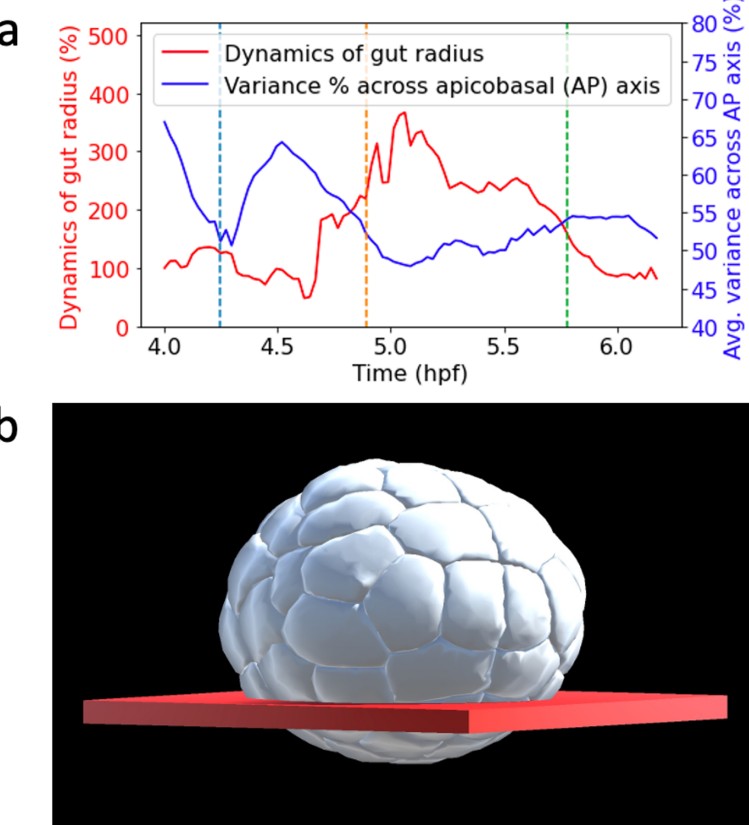

**Appendix 1—figure 7.** Endoderm dynamics. (**a**) Spatial region of point cloud used to observe gut radius dynamics. (**b**) Plot of different endoderm dynamics. Blue plot: average variance of endoderm cells material particle positions across apicabasal axis during gastrulation. Red plot: dynamics of endoderm gut radius.

