## [Editor Report · eLife Assessment]

In this **important** work, a quantitative analysis method for three-dimensional morphogenetic processes during embryonic development is introduced. The proposed method is a pipeline combining several methods, allowing quantitative analysis of developmental processes without cell segmentation and tracking. Upon application of their method, the authors obtain **convincing** evidence that ascidian gastrulation is a two-step process. This work should be of interest to a broad range of developmental biologists who aim to obtain a quantitative understanding of morphogenesis.

---

## [Referee Report · Reviewer #1 (Public review)]

Summary:

The authors propose a new method to quantitatively assess morphogenetic processes during organismal development. They apply their method to ascidian morphogenesis and thus find that gastrulation is a two-step process.

The method applies to morphogenetic changes of surfaces. It consists of the following steps: first, surface deformations are quantified based on microscopy images without requiring cellular segmentation and tracking. This is achieved by mapping, at each time point, a polygonal mesh initially defined on a sphere to the surface of the embryo. The mapped vertices of this polygonal mesh then serve as (Lagrangian) markers for the embryonic surface. From these, one can infer the deformation of the surface, which can be expressed in terms of the strain tensor at each point of the surface. Changes in the strain tensor give the strain rate, which captures the morphogenetic processes. Second, at each time point, the strain rate field is decomposed in terms of spherical harmonics. Finally, the evolution of the weights of the various spherical harmonics in the decomposition is analysed via a wavelet analysis. The authors apply their workflow to ascidian development between 4 and 8.7 hpf. From their analysis they find clear indications for gastrulation and neurulation and identify two sub-phases of gastrulation, namely, endoderm invagination and 'blastophore closure'.

Strengths:

The combination of various tools allows the authors to obtain a quantitative description of the developing embryo without the necessity of identifying fiducial markers. Visual inspection shows that their method works well. Furthermore, this quantification then allows for an unbiased identification of different morphogenetic phases.

Weaknesses:

At times, the explanation of the method is hard to follow, unless the reader is already familiar with concepts like level-set methods or wavelet transforms. Furthermore, the software for performing the determination of Lagrangian markers or the subsequent spectral analysis does not seem to be available to the readers.

---

## [Referee Report · Reviewer #2 (Public review)]

Summary:

In this manuscript, the authors proposed a method to quantitatively analyze 3D live imaging data of early developing embryos, using the ascidian development as an example. For this purpose, the previously proposed level set method was used to computationally track the temporal evolution of reference points introduced on the embryo surface. Then, from the obtained three-dimensional trajectories, the velocity field was obtained, from which the strain rate field was computed. The strain rate field was analyzed using spherical harmonics.

In this paper, the authors focused on the modes with lower order with real coefficients. The time evolution of these modes was analyzed using wavelet transforms. The results obtained by the pipeline reflected the developmental stages of ascidian embryos.

Strengths:

In this way, this manuscript proposes a pipeline of analyses combining various methods. The strength of this method lies in its ability to quantitatively analyze the deformation of the entire embryo without the requirement for cellular segmentation and tracking.

Weaknesses:

The mathematics behind this method is not straightforward to understand. The value of this method will be understood as analyses of real data using this method accumulate.

Comments on revised version:

I have reviewed the revised manuscript and the reply from the authors. All concerns have been addressed appropriately.

---

## [Author Response]

The following is the authors’ response to the original reviews

**Reviewer 1:**
(1) Figure 2 is mentioned before Figure 1

We thank the reviewer for pointing this out, this was a mistake. What was meant by Figure 2 was actually Figure 1. This has been corrected in the manuscript.

(2) Figure 1c: red is used to indicate cell junctions on raw data, but also the error.

The color red is used to indicate cell junctions on raw data on figure 1c left, while it is used to indicate the error on figure 1c right.

The Lagrangian error can be negative right? This is not reflected by the error scale which goes from 0% to 100%

A negative Lagragian error would mean that the distance between real and simulated cellular junctions decreased over time. We effectively treat this case as if there was no displacement, and the error is hence 0%.

Why do you measure the error in percent?

The error is measured in percentages because it is relative to the apical length of a cell.

(3) Figure 2: The distinction between pink and red in e_2(t) is very difficult. What do the lines indicate?

The lines indicate directions of the eigen vectors of the strain rate tensor at every material particle of the embryo.

(4) L156 "per unit length": Rather per unit time?

We thank the reviewer for pointing this out. We apologize for this mistake. "per unit length" has been changed to "per unit time"

(5) L159 "Eigen vectors in this sense": is there another sense?

"In this sense" is referring to the geometric description of eigen vectors. The phrase has been removed

(6) L164 "magnitude of the rate of change underwent by a particle at the surface of the embryo in the three orthogonal spatial directions of most significant rate of change."

Would a decomposition in two directions within the surface's tangent plane and one perpendicular to it not be better?

We also performed the decomposition of the strain rate tensor as suggested within the surface's tangent plane and one perpendicular to it, but did not notice any tangible differences in the overall analysis, especially after derivation of the scalar field.

(7) L174 "morphological activity": I think this notion is never defined

By morphological activity we mean any noticeable shape changes

(8) L177: I did not quite understand this part

This part tries to convey that the scalar strain rate field evidences coordinated cell behaviors by highlighting wide regions of red that traverse cell boundaries (e.g. fig.2b, $t=5.48hpb$). At the same time, the strain rate field preserves cell boundaries, highlighted by bands of red at cellular intersections, when cell coordinated cell behaviors are not preponderant (e.g. fig.2b, $t=4hpb$).

(9) Ll 194 "Unsurprisingly, these functions play an important role in many branches of science including quantum mechanics and geophysics Knaack and Stenflo (2005); Dahlen and Tromp (2021)." Does this really help in understanding spherical harmonics?

This comment was made with the aim of showing to the reader that Spherical Harmonics have proved to be useful in other fields. Although it does not help in understanding spherical harmonics, it establishes that they can be effective.

(10) Figure 3a: I do not find this panel particularly helpful. What does the color indicate? What are the prefactors of the spherical harmonics?

This panel showcases the restriction of the strain rate scalar field to the spherical harmonics with the l and m specified. Each material particle of the embryo surface at the time is colored with respect to the value of \begin{document}$f_{l m}(t) Y_{l m}$\end{document}. The values \begin{document}$f_{l m}(t)$\end{document} are computed according to equation 2 and are showcased in figure 3c.

(11) L 265: Please define "scalogram" as opposed to a spectrogram.

Scalograms are the result of wavelet transforms applied to a signal. Although spectrogram can specifically refer to the spectrum of frequencies resulting for example from a Fourier transform, the term can also be used in a broader sense to designate any time-frequency representation. In the context of this paper, we used it interchangeably with scalogram. We have changed all occurrences of spectrogram to scalogram in the revised manuscript.

(12) L 299 "the analysis was carried out the 64-cell stage.": Probably 'the analysis was carried out at the 64-cell stage'

We thank the reviewer for pointing this out. The manuscript was revised to reflect the suggested change.

(13) L 340 "Another outstanding advantage over traditional is": Something seems to be missing in this sentence.

We thank the reviewer for pointing this out. We have modified the sentence in the revised manuscript. It now reads “Another outstanding advantage of our workflow over traditional methods is that our workflow is able to compress the story of the development ... ”.

(14) Ll 357 "on the one hand, the overall spatial resolution of the raw data, on the other hand, the induced computational complexity.": Is there something missing in this sentence

The sentence tries to convey the idea that in implementing our method, there is a comprise to be made between the choice of the number of particles on the constructed mesh and the computational complexity induced by this choice. There is also a comprise to be made between this choice of the number of particles and the spatial resolution of the original dataset.

**Reviewer 2:**
(1) The authors should clearly state to which data this method has been applied in this paper. Also, to what kind of data can this method be applied? For instance, should the embryo surface be segmented?

The method has been applied on 3D+time imaging data of ascidian embryonic development data hosted on the morphonet (morphonet.org) platform. The data on the morphonet platform comes in two formats: closed surface meshes of segmented cells spatially organized into the embryo, and 3D voxelated images of the embryo. The method was first designed for the former format and then extended to the later. There is no requirement for the embryo surface to be segmented.

(2) In this paper, it is essential to understand the way that the authors introduced the Lagrangian markers on the surface of the embryo. However, understanding the method solely based on the description in the main text was difficult. I recommend providing a detailed explanation of the methodology including equations in the main text for clarity.

We believe that adding mathematical details of the method into the text will cloud the text and make it more difficult to understand. Interested readers can refer to the supplementary material for detailed explanation of the method.

(3) In eq.(1) of the supplementary information, d(x,S_2(t)) could be a distance function between S_1 and S_2 although it was not stated. How was the distance function between the surfaces defined?

What was meant here was *d(x,S_1(t))* where x is a point of S_2(t). *d(x,S_1(t))* referring to the distance between point x and *S_1(t).* The definition of the distance function has been clarified in the supplementary information.

(4) In the section on the level set scheme of supplementary information, the derivation of eq.(4) from eq.(3) was not clear.

We added an intermediary equation for clarification.

(5) Why is a reference shape S_1(0) absent at t=0?

A reference shape *S_1(0)* is absent at *t=0* precisely because that is what we are trying to achieve: construct an evolving Lagrangian surface *S_2(t)* matching *S_1(t)* at all times.

(6) In Figure 2(a), it is unclear what was plotted. What do the colors mean? A color bar should be provided.

The caption of the figure describes the colors: “(a) Heatmap of the eigenvector fields of the strain rate tensor. Each row represents a vector field distinguished by a distinct root color (\begin{document}$\textit{yellow, pink, white}$\end{document}). The gradient from the root color to red represents increasing magnitudes of the strain rate tensor.”

(7) With an appropriate transformation, it would be possible to create a 2D map from a 3D representation shown in for instance Figure 2. Such a 2D representation would be more tractable for looking at the overall activities.

We thank the reviewer for pointing this out. In Figure 4b of the supplementary information, we provide a 2D projection of the scalar strain rate field.

(8) The strain rate is a second-order tensor that contains rich information. In this paper, the information in the tensor has been compressed into a scalar field by taking the square root of the sum of the squares of the eigenvalues. However, such a representation may not distinguish important events such as stretching and compression of the tissue. The authors should provide appropriate arguments regarding the limitations of this analysis.

The tensor form of the strain rate field is indeed endowed with more information than the scalar eigen value field derived. However, our objective in this project was not to exhaust the richness of the strain rate tensor field but rather to serve as a proof of concept that our global approach to studying morphogenesis could in fact unveil sufficiently rich information on the dynamical processes at play. Although not in the scope of this project, a more thorough exploration of the strain rate tensor field could be the object of future investigations.

(9) The authors claimed that similarities emerge between the spatiotemporal distribution of morphogenesis processes in the previous works and the heatmaps in this work. Some concrete data should be provided to support this claim.

All claims have been backed with references to previous works. For instances, looking at figure 2b, the two middle panels on the lower row (5.48hpf, 6.97hpf), we explained that the concentration of red refers respectively to endoderm invagination during gastrulation, and zippering during neurulation [we cited Hashimoto et al. (2015)]. Here, we relied on eye observation to spot the similarities. The rest of the paper provides substantial and robust additional support for these claims using spectral decomposition in space and time.

(10) The authors also claimed that "A notable by-product of this scalar field is the evidencing of the duality of the embryo as both a sum of parts constituted of cells and an emerging entity in itself: the strain rate field clearly discriminates between spatiotemporal locations where isolated single cell behaviours are preponderant and those where coordinated cell behaviours dominate." The authors should provide specific examples and analysis to support this argument.

Here, we relied on eye observation to make this claim. This whole section of the paper “Strain rate field describes ascidian morphogenesis” was about computing, plot and observing the strain rate field.

However, specific examples were provided. This paragraph was building towards this statement, and the evidence was scattered through the paragraph. We have now revised the sentence to ensure that we highlight specific examples:

“A notable by-product of this scalar field is the evidencing of the duality of the embryo as both a sum of parts constituted of cells and an emerging entity in itself: the strain rate field clearly discriminates between spatiotemporal locations where isolated single cell behaviours are preponderant (e.g. fig.2b, $t=4hpb$) and those where coordinated cell behaviours dominate (e.g. fig.2b, $t=5.48hpb$).”

(11) The authors should provide the details of the analysis method used in Figure 3b, including relevant equations. In particular, it would be helpful to clarify the differences that cause the observed differences between Figure 3b and Figure 3c.

Figure 3b was introduced with the sentence: “In analogy to Principal Components Analysis, we measure the average variance ratio over time of each harmonic with respect to the original signal (Fig.3b).” explaining the origin of variance ratio values used in figure 3b. We have now added the mathematical expression to further clarify.

(12) The authors found that the variance ratio of Y_00 was 64.4%. Y_00 is a sphere, indicating that most of the activity can be explained by a uniform activity. Which actual biological process explains this symmetrical activity?

The reviewer makes a good point which also gave us a lot to think about during the analysis. Observing that the contribution of Y00 peaks during synchronous divisions, which are interestingly restricted only to the animal pole, we conjecture that localized morphological ripples and can be felt throughout the embryo.

(13) The contribution of other spherical harmonics than Y_00 and Y_10 should be shown.

Other spherical harmonics contributed individual to less than 1% and we did not find it important to include them in the main figure. We will add supplementary material.